# SMART: Self-learning Meta-strategy Agent for Reasoning Tasks

## Abstract

Tasks requiring deductive reasoning, especially those involving multiple steps, often demand adaptive strategies such as intermediate generation of rationales or programs, as no single approach is universally optimal. While Language Models (LMs) can enhance their outputs through iterative self-refinement and strategy adjustments, they frequently fail to apply the most effective strategy in their first attempt. This inefficiency raises the question: *Can LMs learn to select the optimal strategy in the first attempt, without a need for refinement?* To address this challenge, we introduce SMART (**S**elf-learning **M**eta-strategy **A**gent for **R**easoning **T**asks), a novel framework that enables LMs to autonomously learn and select the most effective strategies for various reasoning tasks. We model the strategy selection process as a *Markov Decision Process* and leverage reinforcement learning-driven continuous self-improvement to allow the model to find the suitable strategy to solve a given task. Unlike traditional self-refinement methods that rely on multiple inference passes or external feedback, SMART allows an LM to internalize the outcomes of its own reasoning processes and adjust its strategy accordingly, aiming for correct solutions on the first attempt. Our experiments across various reasoning datasets and with different model architectures demonstrate that SMART significantly enhances the ability of models to choose optimal strategies without external guidance (+15 points on the GSM8K dataset). By achieving higher accuracy with a single inference pass, SMART not only improves performance but also reduces computational costs for refinement-based strategies, paving the way for more efficient and intelligent reasoning in LMs.

## 1 Introduction

When people first encounter complex reasoning tasks, such as solving mathematical problems, they often make mistakes or approach them inefficiently (Brown et al., 2019). However, with experience, humans tend to improve their performance by replacing ineffective or incorrect strategies with more effective ones, using a mix of strategies tailored to the specific task (Adolph et al., 1998; Lemaire & Callies, 2009; Torbeyns et al., 2009; Boncoddo et al., 2010, *inter alia*).

Language Models (LMs) similarly struggle with reasoning tasks, sometimes producing incoherent results (Madaan et al., 2023; Shridhar et al., 2023; Huang et al., 2024). A common remedy is to resample the output, a process known as *refinement*. This refinement may involve reusing the same reasoning approach (Madaan et al., 2023) or adopting an entirely new one (Shridhar et al., 2023). In addition, providing feedback on initial results has proven beneficial during resampling (Huang et al., 2024; Welleck et al., 2022; Shinn et al., 2024; Kim et al., 2024, *inter alia*). This raises a critical question: *Can LMs be taught to optimize their choice of reasoning strategy for specific tasks overtime on the first trial, much like humans do?*

To address this question, we propose a novel framework called SMART (**S**elf-learning **M**eta-strategy **A**gent for **R**easoning **T**asks), which allows LMs to learn optimal strategy selection through a continuous self-learning approach. We model the task of identifying the optimal strategy as a *Markov Decision Process* (MDP) (Sutton et al., 1999; Puterman, 2014), where the agent (LM) starts with its pre-trained knowledge and iteratively improves its performance by learning from its own outputs and strategy choices. By integrating the LM's reasoning abilities with reinforcement learning-driven

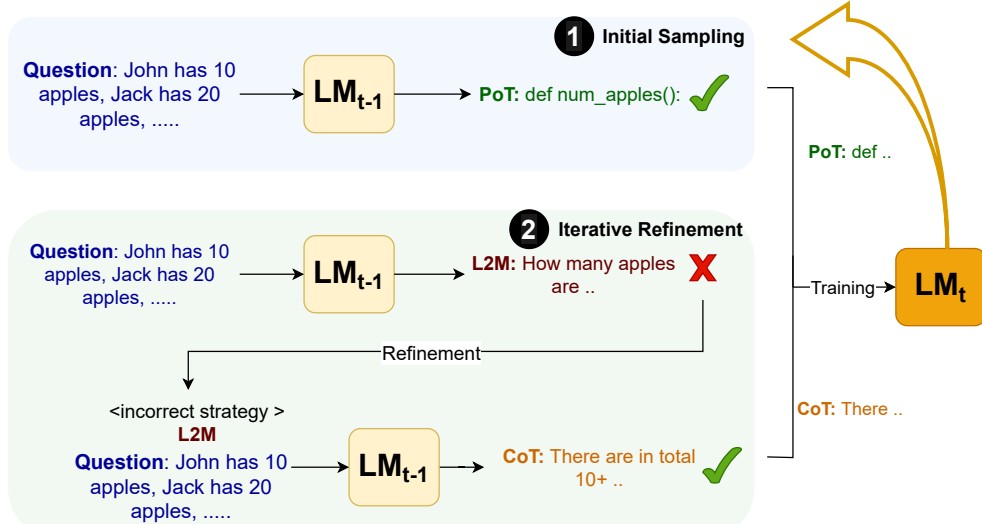

Figure 1: **Our proposed methodology**: In the first step (initial sampling), an agent (LM) chooses a strategy and solves the given task with it. If it is correct, the process ends successfully. If an incorrect strategy is chosen, the agent iteratively refines its strategy, taking previous strategies into account. The process stops when a correct strategy is chosen to solve a task, or when a stopping criterion such as the number of attempts is reached. All correct strategies are used to further refine the model, and the process is repeated. During testing, we sample once from $LM_t$ without refinement.

self-improvement, the agent can simulate different reasoning strategies, evaluate their effectiveness based on past outcomes, and adjust its strategy choice accordingly.

Our approach differs from traditional methods by focusing on iterative reward-based learning, which encourages the agent to produce the correct inference on the first attempt without resampling. This not only improves cost efficiency - only one sampling step is required during inference - but also results in a more generalizable model capable of adapting its strategy selection based on the specific task. We validate SMART on a variety of reasoning datasets and LM architectures and show that our method significantly improves the ability of LMs to select optimal strategies on the first try, outperforming baseline models that rely on traditional self-refinement techniques in both accuracy and computational efficiency. On three mathematical datasets (GSM8K Cobbe et al. (2021), SVAMP Patel et al. (2021b), ASDiv Miao et al. (2020)) over three LLM agents (Llama3 8B Dubey et al. (2024), Gemma 7B Team et al. (2024), and Mistral 7B Jiang et al. (2023)), we demonstrate the effectiveness of our approach. On iterative refinement with SMART, we achieve gains of up to +15 points (a relative gain of +35%) on the GSM8K dataset without the need for refinement. In addition, we improve refinement accuracy by +16 points over baselines.

## 2 METHODOLOGY

Let $q$ be a problem best solved with multi-step reasoning. An example is presented in the form of a mathematical word problem in Figure 1. An agent or language model (LM) can approach it using various strategies, such as solving it step by step (Chain of Thought, CoT Wei et al. (2022)), decomposing it into subproblems, and solving each one (Least to Most, L2M Zhou et al. (2023)), or writing a program to solve it programmatically (Program of Thought, PoT Chen et al. (2023)), among others. A common method is to prompt the LM with a specific strategy for solving the task. However, LMs are error-prone but can fix their answers when asked to do so, a process called *refinement*. In the refinement process, LMs can either stick to the same strategy Madaan et al. (2023) or switch to a more effective reasoning strategy Shridhar et al. (2023). Ideally, LMs could

learn to choose the best strategy and reasoning path on the first try, minimizing the need for costly refinement.

**Our objective:** The primary goal of our work is to enable language models (LMs) to autonomously learn and select the most effective strategies for various reasoning tasks on their first attempt, thereby improving both efficiency and accuracy. Unlike traditional self-refinement methods that require multiple inference passes or external feedback, our approach aims to internalize the learning process within the LM, allowing it to adjust its strategy selection based on past experience. This mirrors how humans learn to choose optimal strategies through experience when faced with complex tasks.

## 2.1 SMART: Self-learning Meta-strategy Agent for Reasoning Tasks

We model the strategy selection process as a *Markov Decision Process* (MDP), where the LM acts as an agent that interacts with the environment (the reasoning tasks) by selecting strategies and observing the outcomes. Using reinforcement learning techniques, the LM can learn a policy that maximizes the expected reward, effectively learning to choose the optimal strategy for each task. This framework allows the LM to simulate different reasoning strategies, evaluate their effectiveness based on past outcomes, and adjust its strategy choice accordingly.

In the following sections, we formalize the problem formulation, define the agent's policy, and describe the learning objective. We then present our two-stage process with iterative refinement, which allows the agent to learn from its own reasoning processes and improve its strategy choice over time.

**MDP Setup:** We model the strategy selection framework as a Markov Decision Process (MDP) given by the tuple $\langle \mathcal{S}, \mathcal{A}, \mathcal{P}, \mathcal{R}, \mu \rangle$. Here, $\mathcal{S}$ represents the state space encapsulating all possible states of the environment, where each state $s \in \mathcal{S}$ is the current problem statement or the subsequent LM response to it. The initial state distribution is given by $\mu$. The action space, $\mathcal{A}$, is the set of all strategies available to the agent, where each action $a_t \in \mathcal{A}$ corresponds to the choice of a particular strategy at time $t$. The transition function $\mathcal{P} : \mathcal{S} \times \mathcal{A} \to \mathcal{S}$ defines the probability of transitioning to a new state $s_{t+1}$ after applying strategy $a_t$ in state $s_t$. Particularly for our case, the transition function is non-deterministic as the next state is sampled by the agent (see Algorithm 1 later). The reward function $\mathcal{R} : \mathcal{S} \times \mathcal{A} \to \mathbb{R}$ assigns a scalar reward based on the correctness of the result after applying the chosen strategy.

We start with the initial sampling step, where the LM chooses a strategy to solve the given task. In other words, given a problem statement $s_1 \sim \mu$, the agent draws a strategy $a_1$ to solve the problem according to its policy (see below). Given the initial problem $s_1$ and strategy $a_1$, the environment transitions to the next state $s_2 \sim \mathcal{P}(\cdot|s_1, a_1)$ with transition probability $\mathcal{P}$ and receives a reward $r_1$ indicating the correctness of the response. If a correct strategy is chosen and the LM solves the task using that strategy, the process terminates. Otherwise, the agent chooses another strategy $a_2$ to solve the problem, and the process repeats until the problem is solved. A realization of this stochastic process is a trajectory $\tau = \left( (s_t, a_t, r_t)_{t=1}^{T-1}, s_T \right)$ up to a finite number of steps $T$. We denote a partial trajectory (history up to time $t$) as $\tau_{1:t} = \left( (s_t, a_t, r_t)_{t=1}^{t-1}, s_t \right)$.

**Agent's policy:** We start by defining a history $h_t := \tau_{1:t}$ that includes the past actions up to time $t - 1$ and the observed states up to time $t$. Then, we model the agent using a non-Markovian stochastic policy $\pi_\theta(a_t|h_t)$ parameterized by $\theta$, which models the probability of choosing the action $a_t$ given the history $h_t$:

$$\pi_\theta(a_t|h_t) = \Pr(a_t = a \,|\, h_t; \theta)$$

**Objective function:** We want to optimize the following objective:

$$\theta^\star = \arg\max_\theta J(\pi_\theta) = \arg\max_\theta \mathbb{E}_{s_1, a_1 \sim \mu_{\text{test}}, \pi_\theta(a_1|s_1)}[r_1(s_1, a_1)] \tag{1}$$

where, $\mu_{\text{test}}$ represents the state distribution over the test data.

TWO-STAGE PROCESS WITH REFINEMENT

STAGE 1 (INITIAL SAMPLING)

1. The process begins with a problem statement $s_1 \sim \mu$ where $\mu$ represents the initial state distribution.

2. The agent samples an action $a_1$ from the policy $\pi_\theta(a_1|h_1)$, where $h_1 = s_1$ for the step 1.

3. The agent then generates an output based on strategy $a_1$

4. The agent receives reward $r_1(h_1, a_1)$ based on it correctness as follows:

$$r_1 = \begin{cases} 1, & \text{if the output is correct} \\ 0, & \text{otherwise} \end{cases}$$

5. **Termination Check:** If $r_1 = 1$, the process terminates successfully.

STAGE 2 (ITERATIVE REFINEMENT, IF $r_1 = 0$)

**For each time step $t = 2$ to $T$:**

1. The agent observes history $h_t$.

2. Samples an action $a_t \sim \pi_\theta(a_t|h_t)$.

3. Generates an output based on strategy $a_t$.

4. Receives reward $r_t(s_t, a_t)$ :

$$r_t = \begin{cases} 1, & \text{if the output is correct} \\ 0, & \text{otherwise} \end{cases}$$

5. **Termination Check:** If $r_t = 1$, terminate the process.

6. Otherwise, transition to the next state $s_{t+1}$, which includes the reasoning choice from all the previous steps.

**Trajectory and Reward Structure:** The resulting trajectory $\tau$ with reward has the following structure:

$$\tau = ((s_1, a_1, r_1), (s_2, a_2, r_2), \ldots, (s_{T'}, a_{T'}, r_{T'}), s_{T'+1})$$

where $T' < T$ is the time step at which the process terminates (either by solving the problem or by reaching the maximum steps). We keep the number of trajectories one less than the number of strategies in our work, since each time the model samples with a different strategy than the one present in its history. For simplicity, we omit $r_t = 0$ rewards in the trajectory above.

The total reward for the trajectory is:

$$R(\tau) = \sum_{t=1}^{T'} r_t$$

Given that $r_t = 0$ for $t < T'$ and $r_{T'} = 1$, if successful, total reward would be $R(\tau) = 1$. However, in some cases, when a task is never solved, the total reward for that trajectory can be 0.

As typical in RL, the agent aims to learn a policy that maximizes the expected cumulative reward (Sutton et al., 1999; Prajapat et al., 2024):

$$\theta^\star = \arg\max_\theta J(\pi_\theta) = \arg\max_\theta \mathbb{E}_{\tau \sim f(\tau; \pi_\theta)}[R(\tau)] \tag{2}$$

POLICY GRADIENT UPDATE

The gradient of the expected reward with respect to the policy parameters $\theta$ is:

$$\nabla_\theta J(\pi_\theta) = \mathbb{E}_\tau \left[ \sum_{t=1}^{|\tau|-1} \nabla_\theta \log \pi_\theta(a_t|h_t) \cdot r_t \right] \tag{3}$$

The gradient is computed iteratively, and a step is only taken at time $t$ only if all the previous attempts were incorrect.

**Implicit bias for Stage 1 to choose the right strategy in the first attempt:** As described in equation 1, our goal is to maximize the reward received as early as possible. To implicitly bias the model towards selecting the correct action in earlier steps, we adjust the dataset $\mathcal{D}$ by replacing the sequences of unsuccessful actions with the final successful action taken at the initial state. Specifically, for trajectories where the problem is solved at time $T'$ ($T' > 1$), we replace the samples $(h_1, a_1, r_1), \ldots, (h_{T'}, a_{T'}, r_{T'})$ with $(h_1, a_{T'}, r_{T'})$. This encourages the model to learn to take the correct strategy $a_{T'}$ for the problem statement $s_1$ in the first step.

Since we only update the model based on correct outputs, the policy update can be viewed as maximizing the likelihood of the correct actions given the history:

$$\theta^{\star} = \arg\max_{\theta} \sum_{(h_i, a_i) \in D} \log \pi_\theta(a_i | h_i) \tag{4}$$

See 1 for a complete algorithm that illustrates our methodology in detail.

## ALGORITHM: SMART: SELF-LEARNING META-STRATEGY AGENT FOR REASONING TASKS

---

**Algorithm 1** Training Procedure for SMART

---

**Require:** Initialized policy parameters $\theta$, learning rate $\alpha$, dataset of problems $\mathcal{D}$.
1: **for** each iteration $e = 1, 2, \ldots$ **do** ▷ For a fixed number of iterations or until convergence
2:      $\mathcal{D}_e \leftarrow \emptyset$          ▷ Initialize empty dataset
3:      **for** each problem $s_1 \in \mathcal{D}$ **do**
4:          Stage 1: Initial Attempt
5:          Observe initial state $s_1$
6:          Sample action $a_1 \sim \pi_\theta(a \mid s_1)$
7:          Generate output using strategy $a_1$
8:          Evaluate output to obtain reward $r_1$
9:          **if** $r_1 = 1$ **then**          ▷ Correct output
10:             Collect sample $(s_1, a_1, r_1)$
11:             $\mathcal{D}_e \leftarrow \mathcal{D}_e \cup \{(s_1, a_1)\}$      ▷ Add sample to dataset
12:             **Continue** to next problem
13:          **else**
14:             Stage 2: Iterative Refinement
15:             **for** each refinement iteration $t = 2 \ldots T$ **do**
16:                 Observe state $s_t$ and history $h_t$
17:                 Sample action $a_t \sim \pi_\theta(a | h_t)$.
18:                 Generate output using strategy $a_t$
19:                 Evaluate output to obtain reward $r_t$
20:                 **if** $r_t = 1$ **then**      ▷ Correct output
21:                    Collect sample $(s_1, a_1, \ldots s_t, a_t, r_t)$
22:                    $\mathcal{D}_e \leftarrow \mathcal{D}_e \cup \{(s_1, a_1, \ldots s_t, a_t)\}$      ▷ Add sample to dataset
23:                    **Break** loop and proceed to next problem
24:                 **end if**
25:             **end for**
26:          **end if**
27:      **end for**
28:      **Policy Update**
29:      Update policy parameters $\theta$ using collected data:

$$\theta \leftarrow \theta + \alpha \cdot \sum_{(h_i, a_i) \in \mathcal{D}_e} \nabla_\theta \log \pi_\theta(a_i | h_i)$$

30:      **Implicit Biasing**
31:      $\mathcal{D}_{e+1} \leftarrow \mathcal{D}_e \backslash \{(s_1, a_1, \ldots s_t, a_t)\} \bigcup \{(s_1, a_t)\}$      ▷ Update the dataset
32: **end for**

---

## 3 EXPERIMENTAL DETAILS

SMART operates within a *self-learn* framework, where we prompt a pre-trained model with 8-shot examples to collect the initial training data. Prompts used for various strategies are provided in Subsection 8.1. The 8-shot pre-trained model also serves as a baseline comparison for our method. We use the MetaMath dataset Yu et al. (2024), a variant of the GSM8K Cobbe et al. (2021) training set, which contains 110K reasoning problems. We evaluate our methodology on the GSM8K test set, which contains 1,319 samples. To show the generalization ability of our method, we also test on two out-of-distribution datasets: the SVAMP dataset Patel et al. (2021a) with 1,000 samples where the questions are made more challenging by altering them in a non-trivial way, and the ASDiv dataset Miao et al. (2020) with 2,300 samples, which consists of diverse mathematical problems from elementary to middle school.

We ran all our experiments on models with 7-8 billion parameters, specifically Gemma 7B Team et al. (2024), Mistral 7B Jiang et al. (2023), Qwen2 7B Yang et al. (2024), and Llama3 8B Dubey et al. (2024). This is important for our *self-learn* setup, as the model needs a basic understanding of the task to begin the process. Smaller models often have difficulty starting the process, while very large models are too expensive to iterate over multiple times.

We employed three reasoning strategies: Chain of Thought (CoT) (Wei et al., 2022), Least to Most (L2M) (Zhou et al., 2023), and Program of Thought (PoT) (Chen et al., 2023) in our work due to their effectiveness on the multi-step reasoning tasks. We take the best out of these strategies as our baseline (underlined in Table 1). Since we used a different reasoning strategy during refinement following Shridhar et al. (2023), the majority of our experiments are limited to two trajectories as the third trajectory would make the remaining strategy the obvious choice. However, we also explore our approach beyond 3 strategies later (in Section 5). We report the top-1 accuracy (maj@1) for all experiments, as we want to test the accuracy of the output on the first try. We used a temperature of 0.7 to generate samples at each iteration. Since the models are already trained on the downstream datasets during pre-training, we chose to train only part of the model and used LoRA (Hu et al., 2021) with rank 16, alpha 32, and a starting learning rate of 2e-4 with decay. We used the Unsloth library (Unslothai, 2023) for training and we did the inference using the VLLM library (Kwon et al., 2023). Finally, we run SMART for multiple iterations until our accuracy gains no longer justify the cost of training and sampling. In our experiments, 3 iterations worked well for most models, although we trained Gemma 7B for 5 iterations before performance plateaued.

## 4 RESULTS

**SMART significantly improves results on in-distribution dataset:** We compared SMART with baselines on the GSM8K dataset, which we also consider to be an in-distribution dataset since the train and test sets have the same distribution. Table 1 shows that SMART outperformed the baseline in its first iteration on the GSM8K dataset, achieving a gain of +6 points for both the Gemma 7B and Mistral 7B models ($40.4 \rightarrow 46.5$ and $56.9 \rightarrow 63.8$, respectively). Although Qwen2's performance is already very strong on the GSM8K dataset, we still observed a gain of +2.6 points ($81.9 \rightarrow 84.5$) in the first iteration. After a few more iterations, we saw a total gain of +15 points for Gemma 7B ($40.4 \rightarrow 55.4$), +11 points for Mistral 7B ($56.9 \rightarrow 67.9$), and +4 points for Qwen2 7B ($81.9 \rightarrow 85.4$).

**SMART serves as a great refinement strategy:** Since SMART involves iterative refinement to find the optimal action given the trajectory, we also compare against two refinement baselines: refinement with the same strategy (Madaan et al., 2023) and refinement with a strategy change (Shridhar et al., 2023). We used the Oracle Verifier, which identifies incorrect samples and refines them either with the same strategy or by choosing a different one. [1] Table 1 compares the refinement accuracy with SMART and our proposed methodology shows significant improvements over the baselines. Gemma 7B gains over +16 points ($48.9 \rightarrow 67.5$) compared to the best refinement baseline, Mistral 7B gains +8 points ($66.5 \rightarrow 78.0$), and Qwen2 7B gains +1.5 points ($86.9 \rightarrow 91.9$).

**SMART generalizes well to out-of-distribution dataset:** We test our trained checkpoints using our proposed approach SMART on two out-of-distribution datasets: ASDiv and SVAMP. These are

---

[1]Note that the Oracle Verifier is used to set the upper bound for all the methods compared and cannot be used during inference. It is only to compare the capabilities of different approaches.

Table 1: Test Accuracy (maj@1) comparison between different baselines using three strategies CoT, L2M, PoT with our approach SMART on the GSM8K dataset. Baselines used 8-shot in-context examples to generate the output. Additionally, we report refinement accuracy obtained through the Oracle verifier for both the baselines and our approach. Refinements for the baselines can follow the *same* strategy as in Madaan et al. (2023) and a *different* strategy than the initial one as in Shridhar et al. (2023). Results are presented for three models: Gemma 7B, Mistral 7B, and Qwen2 7B. The best results among the baseline are underlined while the best overall results are in **bold**.

| Model | Method | Test Accuracy (%) | Refinement | |
| --- | --- | --- | --- | --- |
| | | | Strategy | Accuracy (%) |
| | Baseline (*Using 8-shot examples*) | | | |
| | Chain of Thought (CoT) | 40.0 | same | 44.6 |
| | | | different | 49.4 |
| | Least to Most (L2M) | 34.9 | same | 43.4 |
| | | | different | 48.7 |
| | Program of Thought (PoT) | 40.4 | same | 46.1 |
| | | | different | 51.3 |
| Gemma 7B | SMART (*Proposed Approach*) | | | |
| | *Iteration 1* | 46.5 | SMART | 64.6 |
| | *Iteration 2* | 50.6 | SMART | 64.0 |
| | *Final Iteration - Iteration 5* | **55.6** (↑ +15.2) | SMART | **67.5** (↑ +16.2) |
| | Baseline (*Using 8-shot examples*) | | | |
| | Chain of Thought (CoT) | 50.6 | same | 59.0 |
| | | | different | 67.5 |
| | Least to Most (L2M) | 52.4 | same | 56.6 |
| | | | different | 67.2 |
| | Program of Thought (PoT) | 56.9 | same | 61.3 |
| | | | different | 70.1 |
| Mistral 7B | SMART (*Proposed Approach*) | | | |
| | *Iteration 1* | 63.8 | SMART | 74.1 |
| | *Iteration 2* | 66.3 | SMART | 76.4 |
| | *Final Iteration - Iteration 3* | **67.9** (↑ +11.0) | SMART | **78.0** (↑ +7.9) |
| | Baseline (*Using 8-shot examples*) | | | |
| | Chain of Thought (CoT) | 81.9 | same | 90.4 |
| | | | different | 89.3 |
| | Least to Most (L2M) | 80.0 | same | 84.9 |
| | | | different | 88.7 |
| | Program of Thought (PoT) | 76.9 | same | 86.0 |
| | | | different | 88.6 |
| Qwen2 7B | SMART (*Proposed Approach*) | | | |
| | *Iteration 1* | 84.5 | SMART | 91.4 |
| | *Iteration 2* | **85.4** (↑ +3.5) | SMART | **91.9** (↑ +1.5) |
| | *Final Iteration - Iteration 3* | 85.2 | SMART | 91.2 |

referred to as out-of-distribution because the model was not trained on these datasets but only evaluated on them. Table 2 compares the results of SMART with the baselines (created using 8-shot in context examples) across the three reasoning strategies: CoT, L2M, and PoT. Among the three models, Gemma 7B showed the most improvement, with a gain of +2.6 points on the SVAMP dataset (65.6 → 68.2) and +2.9 points on the ASDiv dataset (68.0 → 70.9). Mistral 7B, on the other hand, gained +1 point on the ASDiv dataset (71.3 → 72.3) but performed worse on the SVAMP dataset. We suspect that the test dataset for SVAMP is different from GSM8K, and since the model was not optimized for SVAMP-style questions, this could explain the drop in performance. Finally, for the Qwen2 7B model, we observed a modest gain of +1 point across both datasets (89.3 → 90.3 on SVAMP and 89.5 → 90.5 on ASDiv).

Table 2: Test accuracy (maj@1) comparison between different baselines using three strategies CoT, L2M, PoT with our proposed approach SMART on the out-of-domain datasets of SVAMP and ASDiv. Baselines used 8-shot in-context examples to generate the output. Results are presented for three models: Gemma 7B, Mistral 7B, and Qwen2 7B. The best baseline results are underlined, while the best overall results are in **bold**.

| Model | Method | SVAMP Acc (in %) | ASDiv Acc (in %) |
|---|---|---|---|
| | Baseline (*Using 8-shot examples*) | | |
| | Chain of Thought (CoT) | 65.6 | 68.0 |
| | Least to Most (L2M) | 63.4 | 62.7 |
| Gemma 7B | Program of Thought (PoT) | 51.6 | 54.2 |
| | | | |
| | SMART (*Proposed Approach*) | | |
| | *Final Iteration* | **68.2** (↑ +2.6) | **70.9** (↑ +2.9) |
| | Baseline (*Using 8-shot examples*) | | |
| | Chain of Thought (CoT) | 65.8 | 71.3 |
| | Least to Most (L2M) | 67.9 | 67.3 |
| Mistral 7B | Program of Thought (PoT) | **71.7** | 70.1 |
| | | | |
| | SMART (*Proposed Approach*) | | |
| | *Final Iteration* | 70.9 (↓ -0.8) | **72.3** (↑ +1.0) |
| | Baseline (*Using 8-shot examples*) | | |
| | Chain of Thought (CoT) | 89.3 | 89.5 |
| | Least to Most (L2M) | 87.5 | 84.2 |
| Qwen2 7B | Program of Thought (PoT) | 82.8 | 80.8 |
| | | | |
| | SMART (*Proposed Approach*) | | |
| | *Final Iteration* | **90.3** (↑ +1.0) | **90.5** (↑ +1.0) |

## 5 DISCUSSION

**How the strategy distribution changes over iterations:** With SMART, the goal is to select the desired strategy at the first attempt, i.e., over iterations the LM should learn to select the appropriate strategy for a given task. Figure 2 shows the changes in strategy distribution over iterations for the Gemma 7B model on the GSM8K dataset. As indicated by the baseline in Table 1, PoT emerges as the best strategy for the Gemma 7B model, followed by CoT, with L2M being the least effective. A similar pattern is observed in Figure 2, where the model increasingly favors PoT (indicated by the upward trend in the red line) while decreasing its preference for the other two strategies. Correspondingly, the accuracy for PoT improves the most, followed by CoT and L2M, demonstrating that over iterations the model learns to select the optimal strategy for a given task. A qualitative example where over iterations the model learned to choose the right strategy in its first attempt is provided in Figure 4. Initially, the model chose the wrong strategy but was fixed during refinement, and over iterations, the model picked the correct strategy in its initial sampling, without the need for refinement.

**Extending the strategies beyond three strategies in SMART:** We want to test if it is possible to go beyond the three strategies explored in this paper by extending our approach to other strategies. Although we could not find a strategy with performance comparable to those explored in this paper, we introduced a <Unsolvable> tag for questions that the model could not answer during either sampling or refinement attempts and tried to use it as a fourth strategy. However, extending this strategy did not yield promising results due to the limited number of samples in the <Unsolvable> category, as the model was able to solve the task using one of the strategies during refinement. Similarly, we experimented with a <Answer Only> strategy, where the model directly predicts the answer to very simple questions without any intermediate reasoning. As with the <Unsolvable> strategy, the skewed data distribution led to poorer performance.

**Is the strategy selection effective?:** We compare the strategy selection made by SMART during inference with a fixed strategy (CoT) on the GSM8K dataset using the Gemma 7B model, as shown in Table 3. Across all five iterations, the results highlight the importance of selecting the appropriate

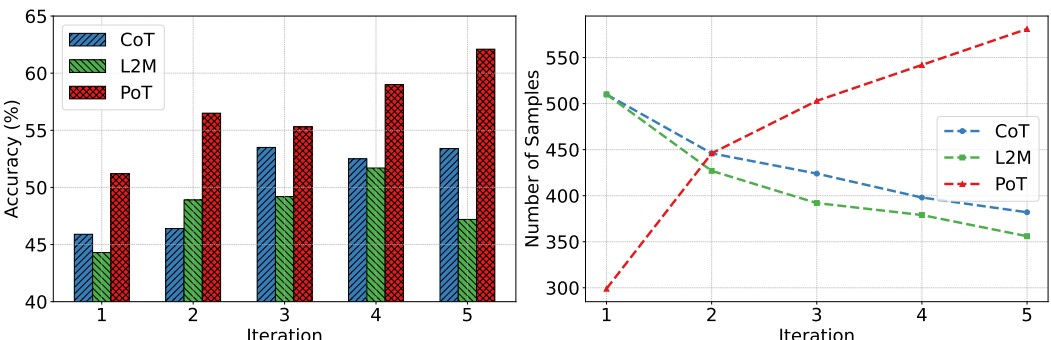

Figure 2: Strategy distribution change over iterations for Gemma 7B model on GSM8K dataset.

Table 3: Comparison of fixed strategy (CoT) vs SMART based strategy during inference for GSM8K dataset using Gemma 7B model.

| Strategy | Iteration 1 | Iteration 2 | Iteration 3 | Iteration 4 | Iteration 5 |
|---|---|---|---|---|---|
| Fixed (CoT) | 45.5 | 47.5 | 48.5 | 50.6 | 49.7 |
| SMART | **46.5** | **50.6** | **52.9** | **55.0** | **55.6** |

strategy, with the SMART approach outperforming the fixed strategy (CoT) by up to 6 points when the optimal strategy is selected. While the fixed strategy shows improvements over iterations, illustrating the *self-learning* effect over multiple generations, our work demonstrates that performance can be further improved by selecting the most appropriate strategy for each task.

**SMART with better starting samples:**

Since we start with the model-generated samples, for a weaker model the starting samples can be improved if those samples come from the stronger model. We investigated whether the SMART based *self-learning* approach could be extended to a setup where the initial data points are collected from a stronger model to initiate training for a weaker model. This is particularly beneficial for weaker models that cannot independently initiate the self-learning process using SMART due to their limited capabilities on a given task. Initially, we collected data using 8-shot in-context examples generated by the Llama3 8B model (Dubey et al., 2024). This approach yielded an average accuracy of about 80% across all three strategies, significantly higher than Gemma 7B's initial accuracy of about

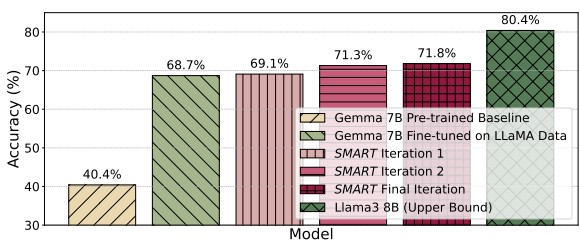

Figure 3: Figure showing a comparison of the effects of different starting data points for the Gemma 7B model. SMART is compared against two baselines: the pre-trained Gemma 7B model and the fine-tuned Gemma 7B model on Llama3 8B data.

40%. Figure 3 illustrates the comparison between different iterations of SMART and the pre-trained Gemma 7B baseline at 40.4%, as well as the fine-tuned Gemma 7B baseline at 68.7%. Using SMART resulted in an additional improvement of +3.1 points (from 68.7% → 71.8%).

## 6 RELATED WORK

**Refinement in LLMs:** Refinement refers to the process of improving the initial output of large language models (LLMs) through iterative adjustments. This refinement can be achieved by following

---

**Question** Brian's friend Bobby has 5 fewer than 3 times as many video games as Brian does. If Brian has 20 video games but lost 5 right before the comparison was made, how many does Bobby have?

**Iteration 1 Stage 1 (Sampling)**
Program of Thought: brian_games = 20
games_lost = 5
bobby_games = 3 * brian_games - games_lost - 5
answer = bobby_games ✗

**Iteration 1 Stage 2 (Refinement)**
Chain of Thought: Brian has 20 video games and lost 5 games, so he has 20 - 5 = 15 games left.
Bobby has 5 fewer than 3 times as many video games as Brian does, so he has 3 * 15 - 5 = 40 video games.
Final Answer: 40 ✓

**Iteration 5 Stage 1 (Sampling)**
Chain of Thought: Brian has 20 - 5 = 15 video games.
Bobby has 5 fewer than 3 times as many video games as Brian does, so Bobby has 3 * 15 - 5 = 40 video games.
Final Answer: 40 ✓

---

Figure 4: Qualitative example demonstrating that Gemma 7B model learnt to refinement strategy in its initial sampling stage, removing the need for refinement.

the same method used initially (Madaan et al., 2023), by incorporating feedback while using the same approach (Welleck et al., 2022; Shinn et al., 2024; Kim et al., 2024; Huang et al., 2024), or by using an alternative method (Shridhar et al., 2023; 2024). However, recent studies have shown that naively applying self-correction can sometimes degrade performance (Huang et al., 2024; Qu et al., 2024; Tyen et al., 2024), highlighting the need for more effective strategies. Supervised fine-tuning with feedback from larger models (Ye et al., 2023; Qu et al., 2024), or using an ensemble of models Havrilla et al. (2024), has produced notable results. Nevertheless, relying on larger or multiple models for feedback presents challenges. In contrast, our approach takes advantage of *self-learning*, eliminating the need for external model feedback.

**Self-Training in LLMs:** Self-training is a semi-supervised learning method in which the model's own predictions are used as additional data to improve its performance (Scudder, 1965; Yarowsky, 1995). This technique has been applied to various NLP tasks, such as machine translation (He et al., 2020; Sun et al., 2021; Gulcehre et al., 2023). In contrast, we use self-learning principles to generate new data and continually update the policy in an on-policy fashion. This approach can be viewed as an on-policy counterpart to Self Imitation Learning (Oh et al., 2018), where the policy learns from prospective successful trajectories in its initial stages, rather than imitating past successful behavior.

## 7 CONCLUSION

We present SMART: Self-learning Meta-strategy Agent for Reasoning Tasks, a solution to the challenges LMs face in selecting strategies for complex reasoning tasks. By modeling the strategy selection process as a Markov decision process and leveraging reinforcement learning, SMART enables LMs to autonomously learn and apply the most effective reasoning strategies on the first trial, thereby reducing the reliance on iterative self-refinement. Our proposed approach not only improves the accuracy of LMs, as demonstrated by significant performance gains across multiple datasets but also enhances computational efficiency by minimizing the need for multiple inference passes.

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

# 8 APPENDIX

## 8.1 PROMPTS

---

Below is an instruction that describes a task, paired with an input that provides further context. Write a response that appropriately completes the request.

Instruction: Solve the given math problem step by step. Put your final answer after 'Final answer:'.

Input: John buys 3 dress shirts. They sell for $20 each. He also has to pay 10% tax on everything. How much did he pay in total?

Response: The shirts cost 3*$20=$<<3*20=60>>60 before tax The tax cost $60*.1=$<<60*.1=6 6 So in total they paid $60+$6=$<<60+6=66 66 Final Answer: 66<eos>

**[7 more examples randomly sampled from the training set]**

Input: Thomas is training at the gym to prepare for a competition. He trained for 5 hours every day for a month (30 days). If he continues to train for the next 12 days, how many hours will he spend on training in total?

Response: Model generated response ..

---

Figure 5: 8-shot Chain of Thought demonstration.

---

Below is an instruction that describes a task, paired with an input that provides further context. Write a response that appropriately completes the request.

Instruction: Solve the given math problem by decomposing it into smaller, manageable sub-questions. Put your final answer after 'Final answer: '.

Input: John buys 3 dress shirts. They sell for $20 each. He also has to pay 10% tax on everything. How much did he pay in total?

Response: Sub-question 1: How much did John spend on shirts?
Answer to Sub-question 1: 3*$20=$<<3*20=60>>60
Sub-question 2: How much did John spend on tax?
Answer to Sub-question 2: $60*.1=$¡¡60*.1=6¿¿6
Sub-question 3: How much did John spend in total?
Answer to Sub-question 3: $60+$6=$<<60+6=66>>66
Final Answer: 66<eos>

**[7 more examples randomly sampled from the training set]**

Input: Thomas is training at the gym to prepare for a competition. He trained for 5 hours every day for a month (30 days). If he continues to train for the next 12 days, how many hours will he spend on training in total?

Response: Model generated response ..

---

Figure 6: 8-shot Least to Most demonstration.

756
757
758
759
760
761
762
763
764
765
766
767
768
769
770
771
772

773 Below is an instruction that describes a task, paired with an input that provides further context. Write a
774 response that appropriately completes the request.

775 Instruction: Solve the given math problem by writing a python program. Store your result as a variable
776 named 'answer'.
777

778 Input: John buys 3 dress shirts. They sell for $20 each. He also has to pay 10% tax on everything.
779 How much did he pay in total?

780 Response: total_shirts = 3
781 cost_of_one_shirt = 20
782 total_cost_shirts = total_shirts * cost_of_one_shirt
783 tax_rate = 0.1
784 tax_amount = tax_rate * total_cost_shirts
785 total_cost = total_cost_shirts + tax_amount
786 answer = total_cost <eos>

787 **[7 more examples randomly sampled from the training set]**
788

789 Input: Thomas is training at the gym to prepare for a competition. He trained for 5 hours every day for a
790 month (30 days). If he continues to train for the next 12 days, how many hours will he spend on training in total?

791 Response: Model generated response ..
792

793 Figure 7: 8-shot Program of Thought demonstration.