# OpenReview forum: "SMART: Self-Learning Meta-strategy Agent for Reasoning Tasks"
_ICLR.cc/2025/Conference — Submitted to ICLR 2025_

### Official Review · Reviewer_XhkN · 2024-10-27

**Soundness:** 2
**Presentation:** 3
**Contribution:** 2
**Rating:** 3
**Confidence:** 3

**Summary:**

The paper tries to solve a problem for language model: Can LMs learn to select the optimal
strategy in the first attempt, without a need for refinement? The strategy I believe indicates the prompt strategy like CoT, PoT, step-by-step, etc. The authors design a strategy policy that selects the strategy based on current state and optimize it to select the best strategy on the first attempt.

**Strengths:**

1.	The topic is interesting since it will reduce the number of attempts that LM needs before achieving successful solution.
2.	Even though there is some details that need to be clarified, the approach is generally easy to understand.

**Weaknesses:**

1.	Authors formulate the problem as MDP, but the policy is designed based on history of states and actions. It conflicts with conventional understanding of MDP in RL field. If the problem has Markov state, there is sufficient to make decision based on state. Otherwise, the state is mistakenly designed and it contains no full information for decision making.
2.	The algorithm is named with “meta-strategy”. People might wonder the meaning of “meta”. Does the policy adjust to different tasks or it just choose appropriate strategy action based on current state of the problem?
3.	The objective is modified to use (4) for policy optimization. Only current outputs are used for update. It becomes a supervised learning with maximum likelihood to reproduce the pattern of correct samples, in stead of reinforcement learning through trial and error.
4.	Following the above question, if the policy is trained on only correct (h_1, a_1/a_t), how it is able to make correct refinement in evaluation where h_t is not seen in the training set. One possible reason I guess might because the authors force “the model samples with a different strategy than the one present in its history” (Line 197). So perhaps randomly selecting the rest strategies in the refinement of SMART may still improve the performance. Authors can refute my guess by adding a comparison experiment.
5.	Does SMART increase the LM parameters and computation because an additional policy LM is needed, in addition to the original pretrained LM to produce problem solutions? If so, I think a discussion is necessary in the paper.

**Questions:**

1.	Equation (1) has a mistake. It is s_1 that samples on \mu, not a_1.
2.	In Algorithm, step 2 makes step 31 totally meaningless. I am also confused if Implicit Biasing should happen before Policy Update or not?

---

> ### Author Response · Authors · 2024-11-20
> **Rebuttal**
>
> We sincerely thank the reviewer for their positive feedback on our work and the clarity of the paper.
> Below, we provide detailed responses to the concerns raised by the reviewer:
>
> ### W1: MDP Formulation
>
> We thank the reviewer for their question. We realized that the reward in our formulation should be defined on the history, specifically, $r : \mathcal{H} \to \mathbb{R}$. We will make this change in line 137.
>
> We agree with the reviewer that if rewards are Markovian, then the state contains complete information to make a decision such that a Markovian policy is optimal.
>
> We would like to point out that the state is a modeling object, and if rewards are non-Markovian (as in our case), then it requires a non-Markovian policy to solve the problem. In our case, the reward depends on both $s_0$ (the initial question) and $s_t$ (the subsequent response to that question). Hence, our policy needs to be conditioned on history to take an effective action.
>
> This is currently an active research area, and there is relevant literature that addresses the expressivity of Markovian rewards [1], examines the sufficiency of Markovian policies for convex rewards [2], and analyzes cases where non-Markovian policies are required [3,4].
>
> [1] https://arxiv.org/abs/2111.00876
> [2] https://arxiv.org/abs/2106.00661
> [3] https://arxiv.org/abs/2307.13372
> [4] https://www.jmlr.org/papers/volume24/22-1514/22-1514.pdf
>
>
>
> ### W2: Meta strategy definition
>
> The term "meta" in our work refers to the process of making a "meta" decision about strategy selection based on the model's past history before solving the problem using the chosen strategy. This is why we used the word “meta” in our work.
>
> ### W3: Difference from supervised learning
>
> **Our proposed methodology cannot be fully addressed using supervised learning alone due to the complexity of unrolling all possible trajectories and capturing the dynamic exploration required for strategy selection**. **SMART enables exploration of strategies across various trajectories and exploits the correct strategies by training on them. This process inherently requires an exploration-exploitation mechanism, which is best achieved using RL.**
>
> For instance, if the model applies a strategy to solve a task but fails, supervised learning methods like rejection sampling can retry with new samples generated. However, these methods do not retain information about the previously failed strategies, which is crucial for the model to understand the need to select a different strategy when a prior one fails. By maintaining a history of attempts, SMART facilitates exploration of alternative strategies and ensures refinement over past actions.
>
> To streamline RL training, we use a simple hard reward system: 0 for incorrect answers and 1 for correct ones. When the model learns the right strategy, we truncate the incorrect trajectory, append the correct one to the dataset (evident from the implicit biasing in line 31 of the presented algorithm), and retrain the model. The process is repeated, allowing the model to explore and learn from all possible trajectories iteratively. This ensures the model not only learns the correct strategy but also develops the ability to apply it efficiently by choosing the right strategy in its first attempt.
>
>
> ### W4: Comparison with baselines for selecting strategies at random
>
> Thank you for raising this excellent point. **We would like to direct the reviewers' attention to the baselines presented in Table 1**. Similar to what the reviewer suggested, **we conducted experiments where the model first samples a strategy and then either resamples using the same strategy (denoted as “same”) or switches to a different strategy randomly** (denoted as “different” in Table 1).
>
> The results show that randomly selecting a different strategy performs worse compared to SMART, which achieves up to a +16-point improvement. For completeness, we repeated the experiments with all strategies used during sampling, iterated over all combinations during refinement, and reported the results in Table 1.
> These findings demonstrate that randomly selecting a new strategy does not yield significant benefits. In contrast, SMART effectively learns to select the most suitable refinement strategy over iterations and subsequently shifts its behavior to choose that strategy on the first attempt. This iterative learning process is key to the superior performance of SMART, as highlighted in the results.
>
>
> ### W5: Additional parameters or computational cost
>
> **The model's parameters are not fully updated during SMART training as we use LoRA** with a rank of 16 and an alpha value of 32, which limits training to approximately 2% of the overall model parameters. **As a result, the computational cost per iteration is relatively low compared to full fine-tuning**.
> We will ensure this detail is clearly articulated in the paper to provide transparency regarding the efficiency and feasibility of our training approach.

---

> > ### Comment · Reviewer_XhkN · 2024-11-24
> >
> > I thank the authors for their discussion.
> >
> > # About W2:
> > I think it is an abuse use of “meta”. Otherwise, any POMDP policy defined on observation history should be called a meta policy.
> > One approach in Meta RL is to formulate the problem as POMDP with the task information not observable, so some algorithms use history of transitions to infer task and design policy. However, it should not be generalized that any policy dependent on history is called a meta policy.
> >
> > # About W3:
> > Authors respond that “However, these methods (supervised learning) do not retain information about the previously failed strategies, which is crucial for the model to understand the need to select a different strategy when a prior one fails. By maintaining a history of attempts, SMART (reinforcement learning) facilitates exploration of alternative strategies and ensures refinement over past actions.”
> > I think this corresponds to Line 196 in the paper: “We keep the number of trajectories one less than the number of strategies in our work, since each time the model samples with a different strategy than the one present in its history.”
> > I wanna make it clear: *is the RL SMART policy able to **actively** select the different strategy than present in history, or it is the same as SL approach that relies on an **add-on mask function** that blocks the options of already selected strategies?*
> > If the former is true, yeah I agree that RL is necessary for that. If the latter is true, then a (RL) policy relying on history and a (SL) predictor relying on history should have no difference.
> >
> > Somehow, I probably understand why authors call this is an RL approach because in RL literature, a considerable researchers think Behavior Cloning belongs to RL. SMART seems to explore (which I think maybe due to the mechanism of selecting different strategies, not RL policy ability) and find a group of successful ones. Then these correct samples are behavior cloned to a policy.
> >
> > Correct me if I think it wrong.
> >
> > # About W4:
> > I thinks the authors now want me to see the comparison in Table 1 between: 1) the results of CoT/L2M/PoT at first step, and followed by a random different strategy; 2) the results of SMART policy at first step, and followed by SMART policy.
> > What I would like to see is the first step uses SMART policy to choose strategy. If failure, the second step randomly chooses a different strategy and see what is the improvement. Then if failure, the third step randomly chooses a different strategy and see what is the improvement. This process repeats until iteration reaches the maximum.
> > The above results are compared with every steps choose strategy according to SMART (that is how results of SMART at Iteration 1, 2, and 5 in Table get, right?).
> >
> > # About W5:
> > My question is to ask if there would be two LM: one to generate response (i.e. think steps and answers to question), the other to select strategy (CoT/L2M/PoT). If true, it will lead to the increase of LM parameters because now you have to keep two LMs at work.

---

> > > ### Author Response · Authors · 2024-11-24
> > > **Reponse II**
> > >
> > > ## W2:
> > >
> > > We acknowledge that the term *meta* might be confusing to some readers, particularly those interpreting it from a meta-learning perspective. In our context, however, **we use *meta-strategy* as a single term referring to strategy selection**. To avoid confusion, we will either use a substitute term or include a footnote clarifying our intended meaning of meta.
> > >
> > > ## W3
> > >
> > > We prompt the model with specific instructions to guide the LLM in selecting a strategy different from the one used in the trajectory. However, **the model retains the option to choose the same strategy if necessary**. This makes it distinct from the add-on mask function, which explicitly blocks the previous choice. In alignment with [1], we encourage the selection of an alternative strategy during refinement, as changing the strategy is generally more beneficial than refining with the same one.
> > >
> > > ## W4
> > >
> > > We understand and appreciate the experimental suggestion. It is worth noting that **in the initial iterations of SMART, the refinement strategy is chosen randomly**, which aligns with the behavior of any RL-based learning approach. As the iterations progress, the model becomes increasingly confident in both Step 1 and Step 2, resulting in iterative improvements over time.
> > >
> > >
> > > ## W5
> > >
> > > We clarify that **we do not use two separate LLMs**. Instead, **the same LLM is employed for both strategy selection and generation**. There is no increase in the model parameters, as both tasks—strategy selection and generation—are performed in an auto-regressive manner.
> > >
> > >
> > > We hope that we have cleared the concerns of the reviewers. We are happy to discuss more.
> > >
> > > [1] https://arxiv.org/abs/2309.13075

---

> ### Author Response · Authors · 2024-11-20
> **Rebuttal II**
>
> ## Questions
>
> ### Q1
> >Equation (1) has a mistake. It is s_1 that samples on \mu, not a_1.
>
> We agree that this is a typo and we will fix that. Thanks for pointing that out.
>
> ### Q2: Implicit biasing before policy update
>
> Our approach prioritizes **updating the policy first, enabling the model to learn which strategy to choose for refinement**. This step is critical to **ensure the model does not make random choices during the exploration phase**. By learning the most effective strategy for refinement through policy updates, the model makes a better decision for various trajectories.
>
> **Once the model has to use some strategy to solve a task, we then want to use that strategy in its first attempt**. To achieve this, we introduce implicit biasing, which biases the model to learn that strategy in its first attempt. The process is then repeated iteratively, allowing the model to refine and solidify its strategy selection over time.
>
> Please let us know if you have any more concerns and we are happy to discuss more.

---

> > ### Comment · Reviewer_XhkN · 2024-11-24
> >
> > About Q2:
> >
> > 1. Authors neglect my first concern about if "In Algorithm, step 2 makes step 31 totally meaningless."
> >
> > 2. As to my second concern, I totally agree that "Implicit Biasing" makes the model learn an effective strategy in its first attempt from $(s_1, a_t, r_t)$.
> > But I didn't understand why we need to learn the failure action $(h_i, a_i)_{i<t}$ from the original sequence $(s_1, a_1, s_2, a_2, \dots, s_t, a_t, r_t)$, which is included by the objective in Algorithm 1, Line 28 "Policy Update".
> > Could you please explain more about that?

---

> > > ### Author Response · Authors · 2024-11-24
> > > **Response**
> > >
> > > We thank the reviewer for the response.
> > > Below are the response to the concerns not clear yet.
> > >
> > > 1. >In Algorithm, step 2 makes step 31 totally meaningless
> > >
> > > Step 2 plays a critical role in refining the model’s approach, as SMART enables the model to adapt its strategy to the given task. If the model selects an incorrect strategy during the first attempt (Step 1), Step 2 allows for refinement by leveraging the history of prior actions. In Step 31, we replace the refined trajectory from Step 2 with the final step of the trajectory. This ensures the model ultimately learns the correct strategy in Step 1 without relying on the refinement process. The combination of Step 2 and Step 31 enables the model to iteratively refine its strategy when errors occur, and ensures that the correct strategy is reinforced for Step 1 in future iterations.
> > >
> > > 2. This is an important concern raised. If we do not make the model learn the right strategy upon failure in step 1, then it will learn to pick the strategy  randomly, which is not ideal. If we compare our work with rejection sampling at step 1, one can allow the model to generate different strategies at random, and reject the incorrect strategies. However, models are shown to be biased to just pick one strategy [1]. With the refinement step 2 in SMART, the model is given a choice to choose another strategy which is hard to pick at random. This way, we can allow the model to learn to explore different strategies for a given task, and eventually learn to pick that in step 1. I hope this makes it clear.
> > >
> > > We are happy to discuss more.
> > >
> > > [1] https://arxiv.org/abs/2410.18574

---

> > > > ### Comment · Reviewer_XhkN · 2024-11-27
> > > >
> > > > ## Algorithm 1
> > > >
> > > > Perhaps we are not talking the same thing :).
> > > >
> > > > When I said about Step 2, I mean the Step number 2 in the Algorithm 1:
> > > >
> > > > - $\mathcal{D}_e \leftarrow \emptyset \qquad$   $\triangleright$ (initialize empty dataset)
> > > >
> > > > which corresponds to the paper line number 240.
> > > >
> > > > The reason I said Step 2 $\mathcal{D}_e \leftarrow \emptyset $ is meaningless, is because when the last $e-1$-th iteration makes the Implicit Biasing at Step 31 by replacing the trajectory data with refined data, at the new $e$-th iteration, Step 2 clears the dataset. So the refined data are added into $\mathcal{D}_e$ but no one has ever used them and they are just removed.

---

> ### Author Response · Authors · 2024-11-27
> **Response to the Reviewer**
>
> This is the updated algorithm:
>
>
>
> ### Algorithm 1: Training Procedure for SMART
>
> **Require:**  Initialized policy parameters θ, learning rate α, dataset of problems D.
>
> 1. **for** each iteration e = 1, 2, . . . **do** ▷ For a fixed number of iterations or until convergence
> 2. &emsp; De ← ∅ ▷ Initialize empty dataset
> 3. &emsp; **for** each problem s1 ∈ D **do**
> 4. &emsp;&emsp; **Stage 1:** Initial Attempt
> 5. &emsp;&emsp; ...
> 6. &emsp;&emsp; **Stage 2:** Iterative Refinement
> 7. &emsp;&emsp; ...
> 8. &emsp; **end for**
> 9. &emsp; **Implicit Biasing**
> 10. &emsp; De+1 ← De\{(s1, a1, . . . st, at)}U{(s1, at)}▷ Update the dataset
> 11. &emsp; **Policy Update**
> 12. &emsp; Update policy parameters θ using collected data
> 13. **end for**
>
> The primary update is that implicit biasing should occur before training the model. This adjustment ensures that our “on-policy” learning approach, as presented in the paper, remains accurate. We hope this explanation resolves any ambiguity—please let us know if further clarification is needed.

---

### Official Review · Reviewer_mfSX · 2024-11-02

**Soundness:** 3
**Presentation:** 2
**Contribution:** 2
**Rating:** 5
**Confidence:** 3

**Summary:**

This paper introduces SMART, a novel framework that enables language models to autonomously learn and select the most effective strategies for reasoning tasks. It models strategy selection as a Markov Decision Process and employs reinforcement learning for continuous self-improvement, aiming to achieve correct solutions on the first attempt. The approach significantly enhances the ability of models to choose optimal strategies without external guidance, improving both accuracy and computational efficiency.

**Strengths:**

see questions

**Weaknesses:**

see questions

**Questions:**

The paper presents an approach to improving reasoning in language models through self-learning strategies. The experimental results seem significant. However, the key details of the method are missing and I cannot fully understand the contribution of this paper.

1. In the methodology section, the authors have emphasized the formulation of MDP, which is a crucial aspect of SMART. However, there is a noticeable absence of critical methodological details. For instance, the refinement process lacks clarity on the specific method (or prompts) used to facilitate the model's self-improvement. Whether additional training conducted to enhance the model's evaluation capabilities? How rewards are derived?  How do data from multiple refinements contribute to improving the model's success rate on the first attempt?

2. In the experimental section, the authors refer to three methods: Chain of Thought (CoT), Least to Most (L2M), and Program of Thought (PoT). How these methods are integrated and utilized within the SMART framework? Why does Table 3 only provide a comparison with CoT and not with the other methods?

---

> ### Author Response · Authors · 2024-11-20
> **Rebuttal**
>
> We thank the reviewer for appreciating the novelty of our framework and valuing our work.
> We address the two questions raised by the reviewer  in detail below:
>
> ### Q1: MDP formulation
> >Method or prompts used for self-improvement
>
> **Our approach employs a two-step process where the model first selects a strategy for a given query and generates a rationale followed by an answer. If the answer is incorrect, the model refines its response by choosing an alternative strategy. This process is repeated until the correct answer is obtained.**
> Once the correct rationale and answer are identified, we use them to train the model to select the correct strategy and generate the correct answer on its first attempt. The prompts used for data generation using different strategies are provided in the appendix for clarity and reproducibility.
>
>
> >How rewards are derived?
>
> **We employ a hard reward system (0-1) for each answer generated within the reasoning trajectories**. A reward of 1 is assigned if the answer is correct, and 0 otherwise. The last reasoning chain with final answer from the trajectories with a reward of 1 are then used to train the model in the subsequent iteration, and this process is repeated iteratively.
> **By using the final rationales and answers from the correct trajectories (reward = 1), we incentivize the model to aim for that reward in its first attempt**. This approach encourages the model to select the correct strategy and generate the correct answer at the outset, improving its performance over iterations.
>
>
> >How do data from refinements leads to models success in its first attempt
>
> **Our approach uniquely focuses on using only the final rationale and answer from the unrolled trajectory to train the model, effectively incentivizing it to select the correct strategy on its first attempt**. Unlike previous methods, which rely on the model to generate an initial (potentially incorrect) answer, decide whether refinement is needed, and then proceed with refinement, our approach seeks to eliminate the need for iterative trial-and-error.
> By training the model with the correct rationale and answer from successful trajectories, we aim to enable the model to internalize and apply the optimal strategy upfront. This distinction underscores our focus on learning to solve tasks correctly on the first attempt, setting our work apart from traditional refinement-based methods.
>
>
>
> ### Q2: Experiments with multiple strategies
>
> The SMART framework enables the model to select any appropriate strategy for a given task. If the chosen strategy is not effective, the model is allowed to select a different one and attempt to solve the task using the new strategy. This approach builds on the demonstrated benefits of heterogeneous strategies over homogeneous ones in refinement tasks, as highlighted by Shridhar et al (https://arxiv.org/abs/2309.13075).  However, **unlike other refinement methods that focus solely on refining outcomes, SMART emphasizes enabling the model to identify and apply the optimal strategy on its first attempt**.
> Figure 2 illustrates this process, showing that **the model initially explores various strategies during training and gradually learns to select the correct one consistently**. This capability is reflected in the significant +15-point improvement on the GSM8K dataset achieved using SMART.
> Table 3 presents an ablation study where the model is allowed to self-learn using any single strategy. For this, we use CoT due to its popularity and proven effectiveness in reasoning tasks. **The study highlights the limitations of self-learning with a fixed strategy, demonstrating that for some tasks, it is critical for the model to adopt alternative strategies. By incorporating multiple strategies, SMART enables the Gemma 7B model to achieve a +5-point gain over self-learning methods that rely on a fixed strategy**. This ability to leverage multiple strategies effectively is one of the core contributions of this work.
>
> Please let us know if you have any more questions and we are happy to discuss more.

---

### Official Review · Reviewer_B8WD · 2024-11-03

**Soundness:** 2
**Presentation:** 3
**Contribution:** 1
**Rating:** 3
**Confidence:** 3

**Summary:**

This paper proposes SMART, a self-training framework that aims to improve problem-solving performance by searching for solutions through self-reflection and subsequently fine-tuning the model to predict the correct solution directly. Specifically, SMART addresses the problem of automatic selection of problem-solving strategies, such as Least-to-Most (L2M) and Chain-of-Thought (CoT), by leveraging reinforcement learning (RL). Empirical results demonstrate that SMART outperforms baseline methods that do not employ self-training.

**Strengths:**

1. **Relevance**: The problem of efficient self-training is important for improving post-training capabilities of large language models (LLMs). A successful self-training method can enhance LLM performance on complex tasks, such as multi-step reasoning, making the research direction highly relevant.
2. **Clarity**: The overall presentation of SMART, including the RL framework for strategy selection, is clearly explained, and the experimental methodology is easy to follow.
3. **Diverse Empirical Evaluation**: Experiments are conducted using several LLM architectures (e.g., Gemma 7B, Mistral 7B) and multiple benchmarks, which adds credibility to the empirical results.

**Weaknesses:**

1. **Limited Novelty**: The core idea of SMART resembles existing self-training methods that utilize rejection sampling (https://arxiv.org/pdf/2308.01825). The concept of generating correct samples through self-reflection has already been widely explored, such as [ReST](https://arxiv.org/pdf/2308.08998) (Gulcehre et al., 2023) and [Re-ReST](https://arxiv.org/pdf/2406.01495), with the only new aspect being the emphasis on strategy selection rather than general solution generation. This difference does not appear significant enough to warrant a new framework.
2. **Lack of Related Work Discussion**: The paper does not adequately discuss related self-training approaches such as [ReST](https://arxiv.org/pdf/2308.08998), [ReST^{EM}](https://arxiv.org/pdf/2312.06585), and other comparable methods like STaR. Including a detailed discussion of these related works would help contextualize SMART’s contributions and clarify how it advances beyond these methods.
3. **Unfair Empirical Comparison**: SMART is compared only against baselines without self-training, which makes the comparisons somewhat unfair. Including direct comparisons against other self-training methods would offer a more meaningful assessment of SMART’s effectiveness.
4. **Contribution of Strategy Selection**: The paper lacks a detailed analysis of the impact of strategy selection. Previous self-training methods, even when limited to a single strategy like CoT, have shown significant improvements. Therefore, it is unclear how much of SMART’s gains can be attributed specifically to strategy selection as opposed to other aspects of the self-training process.
5. **Limitation to LoRA Results**: The experiments only report results from LoRA fine-tuning, which may not fully capture the performance gains that could be achieved by fine-tuning the entire model. While this is acceptable given possible resource constraints, including results from full-model fine-tuning could strengthen the empirical evaluation.

**Questions:**

1. **Comparison with Existing Baselines**: How does SMART compare against other self-training baselines like ReST and ReST^{EM}? A direct comparison with these methods would provide stronger support for SMART’s effectiveness.
2. **Ensuring Strategy Selection**: How does SMART ensure that during data generation, one of the three strategies (L2M, PoT, CoT) is selected? Without proper constraints or instructions, LLMs might struggle to consistently use the intended strategies. Does SMART simply try each of these strategies until a correct solution is found, or is there a more sophisticated mechanism in place?
3. **Algorithm Implementation**: In the algorithm description, should the implicit biasing step occur before the policy update? This order seems more intuitive and would ensure the update incorporates the biased data correctly.

---

> ### Author Response · Authors · 2024-11-20
> **Rebuttal**
>
> We thank the reviewer for appreciating the relevancy of our framework and valuing the clarity of our work.
> We address the weaknesses highlighted by the reviewers, along with the questions raised, in detail below:
>
> ### W1: Comparison with ReST or Rejection Sampling
>
> **We have compared our methodology with ReST both conceptually and through experiments**. It is important to note that while ReST focuses on iteratively improving performance through self-learning for machine translation tasks, it follows a distinct approach. ReST trains a model, generates multiple samples, filters samples above a certain threshold, and iteratively trains the model with those selected samples in an off-policy setting by combining old data with the newly generated data.
> In contrast, our approach and task choice are fundamentally different. We focus on determining the most effective reasoning strategy for solving a task by allowing the model to decide which strategy to use for a given task. **While we share the concept of a self-iterative training loop with rewards, our method enables the model to select a strategy, and if it fails, to choose another strategy conditioned on prior decisions, continuing along this trajectory. There is no concept of refinement present in ReST**. Later, we reinforce the model's ability to pick the correct strategy in the initial step by simulating training, unrolling the trajectory, and identifying the optimal one iteratively in an on-policy setting.
> Moreover, **we have presented the results inspired by ReST style self-learning in Table 3 of the paper. We highlight that self-training using a fixed strategy performs significantly worse than SMART**.
>
> **Rejection sampling**, on the other hand, involves generating multiple reasoning paths using the model itself and selecting the correct ones to augment the training dataset. This approach **is fundamentally different from ours. Instead of simply resampling when a reasoning path is incorrect, our method allows the model to choose a strategy and, if the initial choice is incorrect, select another one conditioned on the prior decisions**.
> Rejection sampling would merely repeat the sampling process in such cases, whereas our approach is more aligned with refinement-based methods. Specifically, we enable the model to iteratively refine its answers based on its past decisions. To benchmark our approach, we have compared it against no refinement, refinement using the same strategy as proposed by Madaan et al. (https://arxiv.org/abs/2303.17651), and refinement using the strategy proposed by Shridhar et al. (https://arxiv.org/abs/2309.13075). This ensures a comprehensive evaluation of our refinement-centric methodology.
>
>
> ### W2: Discussion on past related works
>
> We have discussed ReST in the related work but we will add more details as specified above. We will also compare our work with rejection sampling and similar works as mentioned above.
>
> ### W3: Comparison with self-training baselines
>
> **Since SMART is an iterative refinement framework, we compare it against several baselines: no refinement** (using 8-shot in-context examples), **refinement using the same strategy** as proposed by Madaan et al. (https://arxiv.org/abs/2303.17651), and **refinement using a different strategy** as proposed by Shridhar et al. (https://arxiv.org/abs/2309.13075). For each baseline, we applied the three strategies introduced in this work: CoT, L2M, and PoT. **This results in a total of 3×2  (refinement baselines) and one no-refinement baseline**. We believe these comparisons encompass all relevant baselines, but we welcome feedback if anything specific was overlooked.
> Additionally, Table 3 demonstrates the advantage of SMART over self-iterative methods without strategy chains, one of the key contributions of this work. SMART outperforms these methods by more than +5 points. Importantly, our self-iterative learning approach represents an on-policy adaptation of the ReST framework, further emphasizing the distinction and effectiveness of SMART.
>
>
> ### W4: Usefulness of strategy selection
>
> We have conducted the exact experiments suggested by the reviewers. **Table 3 specifically compares a self-iterative approach using a fixed strategy, CoT, against SMART, which incorporates dynamic strategy selection over time**. As shown in Table 3, **SMART leverages strategy selection effectively, resulting in a +5 points improvement over the self-iterative CoT strategy**. This highlights the benefit of allowing the model to adapt its strategy dynamically, a key strength of SMART.
>
>
> ### W5: Limitation to LoRA results
>
> We restricted our experiments to LoRA because the models used had already been pre-trained on mathematical reasoning tasks, and full fine-tuning resulted in overfitting. **By training only specific parts of the model, LoRA provided an efficient and effective solution, enabling us to achieve meaningful improvements without the risk of overfitting**.

---

> ### Author Response · Authors · 2024-11-20
> **Rebuttal II**
>
> ## Questions
>
> ### Q1: Comparison to past works
>
> Please refer to the answer in Weakness 2 above where we have discussed the comparison of SMART with past refinement methods and an adaptation of ReST suitable for our work.
>
> ### Q2: Ensuring Strategy selection
> We have illustrated the effectiveness of strategy selection over time in Figure 2, which highlights the benefits of dynamically selecting the appropriate strategies as tasks progress. This is achieved through a reward mechanism designed to encourage the model to select the optimal strategy on its first attempt.
> **When SMART simulation begins, it initially picks strategies at random, similar to other reinforcement learning frameworks. However, over time, the reward mechanism incentivizes the model to consistently choose the correct strategy**. This is accomplished by identifying the correct rationale and answer from various trajectories and using them to guide the model in its early decision-making. This process is further supported by the implicit biasing step outlined in line 31 of the Algorithm, reinforcing the model's learning to pick the optimal strategy at the start.
> Finally, **Table 3 also compares strategy selection using SMART vs self-training with one strategy (CoT). SMART outperforms CoT based self-training by over 5 points, demonstrating the usefulness of strategy selection**.
>
> ### Q3: Implicit biasing before policy update in the Algorithm
>
> **We do not apply implicit biasing before the policy update because it is crucial for the model to learn to select the appropriate refinement strategies through iterative exploration.** This approach ensures that the model moves away from choosing random strategies repeatedly and instead focuses on refining its answers based on identified mistakes. Later, we train the model with right reasoning rationales in the implicit biasing step.
>
>
> Please let us know if you have any more concerns and we are happy to discuss more.

---

> > ### Comment · Reviewer_B8WD · 2024-11-27
> >
> > I thank the authors for their clarification, which addresses some of my concerns. However, main concerns remain unaddressed.
> >
> > **Q1:** The authors claim that "highlight that self-training using a fixed strategy performs significantly worse than SMART". However, this comparison is not fair. To demonstrate the effectiveness of SMART, the authors should run self-training methods to also optimize strategy selection, rather than optimizing solution generation under a fixed strategy. I believe both ReST and Re-ReST are suitable for running this evaluation, as the strategy selection problem can also be viewed as a reasoning problem, and correctness signal is easily accessible.
> >
> > **Q2:** How does SMART work at test time? Does it only selects the strategy for once, or selects for multiple times? If it selects for multiple times, does it need to know the correctness signals in previous trails? If so, the comparison in Table 3 may be unfair, as the fixed strategy (CoT) baseline only queries correctness for once.
> >
> > **Q3:** The implicit biasing part is still confusing. In line 31, the implicit biasing is applied to $D_{e+1}$. However, in the next iteration, in line 2, the dataset $D_{e+1}$ is again initialized to an empty set. So according to the algorithm box, this step seems useless. Also, there is no ablation study demonstrating the necessity of implicit biasing.

---

> ### Author Response · Authors · 2024-11-28
> **Response clarifying reviewers concerns**
>
> We thank the reviewer for the discussion. Below, we provide clarifications for the concerns raised.
>
> ## Q1 : Comparison with ReST
> As previously mentioned, ReST employs off-policy training where data from the latest checkpoint is combined with all past data, and the initial checkpoint is retrained with this aggregated data. However, **training with this mixed data did not yield positive results for us**. Our findings align with prior work [1], which reached a similar conclusion (see the "combined" results in Table 1). The effectiveness of SMART lies in its iterative on-policy training, which enables the model to identify suboptimal strategies, refine them, and eventually learn to select an effective strategy on the first attempt. This requires iterative on-policy training to continually learn this. ReST relies on off-policy training which is not ideal for this task.
>
> We hope this explanation clarifies why ReST is not well-suited for this task.
>
> ## Q2: How many time SMART samples during inference?
> **SMART samples only once during inference, ensuring a fair comparison with other methods** in Table 1 and Table 3, which also use single-sample inference. SMART does not employ any additional inference tricks; it performs straightforward auto-regressive decoding. Simply, SMART first selects a strategy and decodes using the chosen strategy in an auto-regressive manner.
>
> ## Q3: Dataset D initialized with empty set
>
> This was a typo on our part, and we clarified this to Reviewer XhkN. We sincerely apologize for this mistake.
>
> **This is a typo as the line number 240 (where the data is initialized) should be done once at the start and not after every epoch. This should be after line and initialized once before the loop starts for each iteration.**
>
> We will make this update in the paper by moving it after line 238.
>
> **Updated algorithm**:
>
> Require: Initialized policy parameters θ, learning rate α, dataset of problems D.
>
> 1: De ← ∅
>
> 2: for each iteration e = 1, 2, . . . do ▷ For a fixed number of iterations or until convergence
>
> ## Update:
> Please ignore the last comment (**Q3: Dataset D initialized with empty set**) and below is the correct algorithm:
>
> ### Algorithm 1: Training Procedure for SMART
>
> **Require:**  Initialized policy parameters θ, learning rate α, dataset of problems D.
>
> 1. **for** each iteration e = 1, 2, . . . **do** ▷ For a fixed number of iterations or until convergence
> 2. &emsp; De ← ∅ ▷ Initialize empty dataset
> 3. &emsp; **for** each problem s1 ∈ D **do**
> 4. &emsp;&emsp; **Stage 1:** Initial Attempt
> 5. &emsp;&emsp; ...
> 6. &emsp;&emsp; **Stage 2:** Iterative Refinement
> 7. &emsp;&emsp; ...
> 8. &emsp; **end for**
> 9. &emsp; **Implicit Biasing**
> 10. &emsp; De+1 ← De\{(s1, a1, . . . st, at)}U{(s1, at)}▷ Update the dataset
> 11. &emsp; **Policy Update**
> 12. &emsp; Update policy parameters θ using collected data
> 13. **end for**
>
>
> Please let us know if your concerns are clear now. We are happy to discuss more.
>
> [1] https://arxiv.org/abs/2410.18574

---

> > ### Comment · Reviewer_B8WD · 2024-12-01
> >
> > Thank you for your response. My concern on Q3 has been addressed. However, I am even more confused about Q1 and Q2.
> >
> > 1. The notions of "on-policy" and "off-policy" are used incorrectly. "On-policy" referes to training on data collected by the current policy only, while "off-policy" refers to training on data collected by both current and previous policies. According to the authors' clarification to Q3, in SMART, the policy is trained on a mixture of data collected by both current and previous policies (as shown in the implicit-bias step, data from previous iterations are adopted to the next iteration), so it is in-adequate to call SMART a "on-policy" algorithm.
> >
> > 2. ReST does not retrain from the initial checkpoint. According to the original paper, it "finetune the current best policy". I think the authors have confused ReST [1] and $ReST^{EM}$ [2].
> >
> > 3. I do not understand why off-policy training completely fails, as SMART also reuses data from previous iterations. Is there an intuitive explanation?
> >
> > 4. Since SMART only selects once during inference, I think an ablation is needed: simply replace SMART's exploration by traversing all three strategies, and train SMART only on successful trails. This should be the (final) performance upper bound for SMART, although it will take more exploration samples than SMART. This ablation will help us better understand how crucial and effective SMART's exploration is. I understand the rebuttal is reaching its end and time may not be sufficient for additional experiments, but I would like to see some discussion on this.
> >
> > References:
> >
> > [1] Gulcehre, Caglar, et al. "Reinforced self-training (rest) for language modeling." arXiv preprint arXiv:2308.08998 (2023).
> >
> > [2] Singh, Avi, et al. "Beyond human data: Scaling self-training for problem-solving with language models." arXiv preprint arXiv:2312.06585 (2023).

---

> ### Author Response · Authors · 2024-12-02
> **Response clearing some confusion**
>
> We appreciate the reviewer’s valuable feedback and would like to address and clarify a misunderstanding that happened because of our last reply:
>
> 1. We align with the definitions of “on-policy” and “off-policy” learning. In our work, **we exclusively use “on-policy” data, meaning that for every sample, the data is collected only from the most recent checkpoint**. This process consists of two stages: stage 1, where the model picks the correct strategy and solution on its first attempt, and stage 2, the refinement stage. Using implicit biasing, we select only the final correct trajectory and train the model with it.
>
> There appears to be some confusion regarding the reply to the initialization of De ← ∅ , which we aim to clarify and simplify within the algorithm.
> **This is our algorithm as per the paper**:
>
> ### Algorithm 1: Training Procedure for SMART
>
> **Require:**  Initialized policy parameters θ, learning rate α, dataset of problems D.
>
> 1. **for** each iteration e = 1, 2, . . . **do** ▷ For a fixed number of iterations or until convergence
> 2. &emsp; De ← ∅ ▷ Initialize empty dataset
> 3. &emsp; **for** each problem s1 ∈ D **do**
> 4. &emsp;&emsp; **Stage 1:** Initial Attempt
> 5. &emsp;&emsp; ...
> 6. &emsp;&emsp; **Stage 2:** Iterative Refinement
> 7. &emsp;&emsp; ...
> 8. &emsp; **end for**
> 9. &emsp; **Implicit Biasing**
> 10. &emsp; De+1 ← De\{(s1, a1, . . . st, at)}U{(s1, at)}▷ Update the dataset
> 11. &emsp; **Policy Update**
> 12. &emsp; Update policy parameters θ using collected data
> 13. **end for**
>
>
> **Major clarification in the algorithm**
> The primary update is that implicit biasing should occur before training the model. This adjustment ensures that our “on-policy” learning approach, as presented in the paper, remains accurate. We hope this explanation resolves any ambiguity—please let us know if further clarification is needed. We apologize for the last reply. We will correct it.
>
> 2. To elaborate on the distinctions, ReST-EM trains from the initial checkpoint, while ReST trains from the most recent checkpoint.  This is correct. **The key difference between these approaches and our work lies in the contrast between off-policy and on-policy learning**.
>
> 3. Off-policy learning does not align well with our framework because, at each step, the model may predict a strategy that could either be correct or incorrect. **While it is possible to select the correct chain from the last checkpoints, multiple valid strategies could emerge for a given problem**. However, prior work [1] demonstrates that such an approach does not work effectively out of the box. **A model may select a strategy from older checkpoints and later switch to a better one in subsequent checkpoints. Retaining the older strategy inhibits this natural evolution, a challenge we observed in our experiments as well**.
>
> 4. For evaluation on the test set with different strategies, we fixed all three strategies and sampled their responses. **The upper bound was calculated by taking the union of all strategies, and we compared this with SMART’s ability to dynamically choose the most optimal strategy** using the Gemma 7B model on the GSM8k dataset:
>
> | GSM-Test | Epoch 1 | Epoch 2 | Epoch 3 | Epoch 4 | Epoch 5 |
> |-----------|---------|---------|---------|---------|---------|
> | Upper bound      | 47.9   | 50.9   | 53.8   | 55.2   | 55.8   |
> | SMART      | 46.5  | 50.6   |   52.9 | 55.0 | 55.6 |
>
> **These results demonstrate that SMART effectively learns to select the optimal strategy over successive iterations.**
>
>
> We aplogize for our last comment which created more confusion. We hope that this fixed all of it. We are happy to know your views.
> Thanks again.
>
>
>
>
> [1] https://arxiv.org/abs/2410.18574

---

### Official Review · Reviewer_tH9T · 2024-11-06

**Soundness:** 3
**Presentation:** 3
**Contribution:** 3
**Rating:** 6
**Confidence:** 4

**Summary:**

The paper introduces SMART, a novel reinforcement learning framework enabling language models (LMs) to autonomously select optimal reasoning strategies for complex tasks on the first attempt, reducing reliance on iterative refinements. By modeling strategy selection as a Markov Decision Process, SMART applies continuous self-improvement, enabling LMs to adapt strategies dynamically. Experiments across datasets show significant performance gains and computational efficiency, with SMART achieving up to +15 points in accuracy on the GSM8K dataset, outperforming traditional refinement methods.

**Strengths:**

1. SMART provides a novel framework by modeling strategy selection in LMs as a Markov Decision Process, adding a unique layer of self-learning.
2. The paper demonstrates high-quality experimentation, validating SMART’s effectiveness across multiple datasets and model architectures, such as GSM8K, SVAMP, and ASDiv.
3. The paper  is clearly articulated, particularly in its two-stage process design, making the approach easy to understand and apply.
4. By enabling LMs to select optimal strategies without iterative refinements, the work significantly reduces computational costs.

**Weaknesses:**

1. The method is overly complex without sufficient justification for why simpler methods cannot achieve similar results. There should be a more detailed baseline comparison.

2. The experimental section lacks clarity, particularly regarding the setup and specific configurations for each dataset, making the results difficult to interpret and replicate. Besides, the paper's discussion on the generalization capability of SMART across different datasets is weak, as it does not convincingly demonstrate robustness beyond minor accuracy gains.

3. The use of only 7B models raises concerns about the scalability of the proposed method, with no evidence provided for its effectiveness on larger models.

4. Theoretical claims about strategy optimization are vague and insufficiently supported by quantitative evidence, especially regarding the selection mechanism's consistency across different reasoning tasks.

**Questions:**

1. The method introduced appears intricate, particularly with its reliance on the MDP. Could you provide more concrete evidence or reasoning to support the necessity of this complex setup over simpler alternatives, such as supervised learning with strategy labels or existing reinforcement learning baselines?

2. Experimental details on configurations and hyperparameters are sparse. You should specify the settings for each dataset and model, especially for reward design and training.

3. The generalization to OOD datasets shows limited gains. What factors might limit this, and could SMART be adapted to improve performance on diverse datasets?

4. Only 7B parameter models were tested. Can you scale SMART to larger models, like 70B parameters?

5. The strategy selection process is crucial but lacks detailed evidence of consistency across tasks. Could you provide quantitative results showing its reliability on different reasoning tasks?

**Details Of Ethics Concerns:**

No.

---

> ### Author Response · Authors · 2024-11-20
> **Rebuttal**
>
> We thank the reviewer for recognizing and appreciating our work, as well as for their positive feedback on our methodology, paper writing, and experimentation choice.
>
> We address the weaknesses highlighted by the reviewers, along with the questions raised, in detail below:
>
>
> ### W1: Comparison with simpler baselines
>
> **As SMART is an iterative refinement framework, we compared it against three baselines**: **no refinement** (8-shot in-context examples), **refinement using the same strategy** as proposed by Madaan et al. (https://arxiv.org/abs/2303.17651), and **refinement using a different strategy** as proposed by Shridhar et al. (https://arxiv.org/abs/2309.13075). We applied each of these baselines to the three strategies introduced in this work: CoT, L2M, and PoT. **This results in a total of 3 × 2 (refinement baselines) plus one no-refinement baseline**. We ensured SMART was compared with all relevant baselines. If we have overlooked any, please let us know, though we believe our chosen baselines are highly suitable for the task.
>
> ### W2: Specific setup configurations and generalization beyond accuracy on out of domain datasets
> We have provided extensive experimental details, including the libraries used, learning rate, LoRA configuration, dataset information, temperature, and more (lines 293-300). If any specific information is missing, please let us know.
> **The purpose of training on one dataset and testing on two out-of-domain datasets was to demonstrate that our approach does not overfit to the dataset it was trained on**. We report accuracy gains of +3 points on Gemma 7B and +1 point using the Mistral and Qwen2 7B models when tested on out-of-domain datasets. **These results confirm that our approach generalizes well** and would likely show similar improvements if trained on other datasets. However, the main objective of testing on out-of-domain datasets was not to measure generalization but to demonstrate robustness to dataset variations.
>
> ### W3: Training larger than 7B models
>
> Our approach relies on self-iterative training over **multiple iterations, involving alternating phases of training and sampling**. **Scaling this process beyond 7B models demands substantial computational resources**, which are currently beyond our availability. Therefore, we have restricted our experiments to models up to 7B in size.
>
>
> ### W4: Quantitative evidence that the approach works
>
> We report accuracy on the GSM8K dataset in Table 1, which served as the primary training dataset for our three models. All three models demonstrate notable improvements, with the Gemma 7B **model achieving gains of up to 15 points**. We believe this provides strong qualitative evidence that our approach is effective. Additionally, Table 3 highlights the importance of strategy selection, showing its impact compared to relying on a single strategy (CoT). **SMART outperforms self-training with a fixed strategy by over 5 points**.

---

> ### Author Response · Authors · 2024-11-20
> **Rebuttal II**
>
> ## Questions
>
> ### Q1: Baselines comparison and reliance on MDP
> - We report the baseline comparisons with past refinement works. Please refer to weakness 1 (W1) above for more details.
> - **The use of MDP was essential to illustrate the various trajectories the model can follow**. SMART also highlights the importance of refinement if the initial sampling was incorrect. Later, we prune the suboptimal trajectories by selecting the correct rationales and answers from different paths and training the model to learn them in this first step. This ensures a more efficient and effective refinement process where the model learns to correctly answer a given problem without any need for refinement.
>
> ### Q2: Hyper-parameters configuration and reward design
> - We have provided all the hyperparameters in the paper to replicate our results. Please refer to weakness 2 above and (lines 293-300) in our paper for hyper-parameters details.
> - In terms of reward design, we use a 0-1 (hard reward) to decide if we want to stop the trajectory or do another step of refinement. **Once we get the correct reward of 1, we use that final answer with rationale to train the model to learn the right rationale and answer in the first step**.
>
> ### Q3: Scaling to larger models
> - Although we would like to test the scalability of our approach, we are limited by compute as we need to train and sample the model multiple times, which require a significant amount of resources to scale the approach.
>
> ### Q4: Importance of strategy selection
> Figure 2 demonstrates the importance of strategy selection where the model learns to choose different strategies over iterations while keeping the most accurate one dominant over iterations. This leads to better overall accuracy. Moreover, **Table 3 in the paper compares the self-learning approach** with just one strategy (CoT) vs SMART with different strategy selection. **SMART outperforms self-training with CoT by over 5 points**.
>
>
> Please let us know if you have any more concerns. We are discuss more.

---

### Meta-Review · Area_Chair_KoLt · 2024-12-20

**Metareview:**

This paper, while presenting an interesting framework (SMART) that models strategy selection in LMs as a Markov Decision Process, ultimately falls short of being acceptable for publication.  Although the authors highlight the novelty of their approach, the clarity of their experimentation, and the computational efficiency gains, several significant weaknesses undermine its contribution. The complexity of SMART seems unwarranted without a more rigorous comparison to simpler baselines, raising doubts about its necessity. Further, the experimental section is poorly detailed, hindering both replicability and a clear understanding of the results. The claims of generalization across datasets are weak, with limited evidence of robust performance beyond marginal improvements. Concerns also arise regarding the scalability of the method, given its testing on only 7B models, and the lack of theoretical grounding for some of the central claims. Therefore, despite the paper's strengths in its innovative concept and clear articulation of the two-stage process, the methodological concerns and insufficient empirical support lead to its rejection.  The authors are encouraged to address these issues in future work.

**Additional Comments On Reviewer Discussion:**

Additional concerns have been addressed by the authors during rebuttal but still the paper haven't passed the threshold of acceptance.

---

### Decision · Program_Chairs · 2025-01-22

Reject